# Exploring the Holistic Needs of People Living with Cancer in Care Homes: An Integrative Review

**DOI:** 10.3390/healthcare11243166

**Published:** 2023-12-14

**Authors:** Stephanie Craig, Yanting Cao, James McMahon, Tara Anderson, Patrick Stark, Christine Brown Wilson, Laura Creighton, Silvia Gonella, Laura Bavelaar, Karolina Vlčková, Gary Mitchell

**Affiliations:** 1School of Nursing and Midwifery, Queen’s University Belfast, Belfast BT9 7BL, UK; scraig22@qub.ac.uk (S.C.); j.mcmahon@qub.ac.uk (J.M.); tanderson@qub.ac.uk (T.A.); p.stark@qub.ac.uk (P.S.); c.brownwilson@qub.ac.uk (C.B.W.); laura.creighton@qub.ac.uk (L.C.); 2The Shanghai Medical College, Fudan University, Shanghai 200437, China; yanting.teamleader@stellarcarenw.co.uk; 3Stellar Care NW Ltd., Ellesmere Port CH65 1A, UK; 4Direction of Health Professions, City of Health and Science University Hospital of Torino, Corso Bramante 88-90, 10126 Turin, Italy; silvia.gonella@unito.it; 5Department of Public Health and Pediatrics, University of Torino, Via Santena 5 bis, 10126 Turin, Italy; 6Department of Public Health and Primary Care, Leiden University Medical Center, 2333 ZA Leiden, The Netherlands; 7Center for Palliative Care, 101 00 Prague, Czech Republic; k.vlckova@paliativnicentrum.cz

**Keywords:** cancer, care homes, nursing homes, integrative literature review, holistic needs, holistic care, older people

## Abstract

Up to 26% of individuals residing in care homes are impacted by cancer. This underscores the importance of understanding the holistic needs of care home residents living with cancer to enhance the quality of their care. The primary objective of this integrative literature review was to consolidate the available evidence concerning the comprehensive needs of people living with cancer in care home settings, providing valuable insights into addressing their diverse needs. An integrative literature review was conducted using a systematic approach. Extensive searches were conducted in three databases, complemented by a thorough examination of grey literature and reference lists of relevant papers. The review focused on literature published between 2012 and 2022. The screening process involved two independent reviewers, with a third reviewer resolving any discrepancies. The review identified twenty research papers that met the eligibility criteria. These papers shed light on three primary themes related to the holistic needs of care home residents with cancer: physical, psychological, and end-of-life needs. Physical needs encompassed pain management, symptom control, and nutrition, while psychological needs involved social support, emotional well-being, and mental health care. End-of-life needs addressed end-of-life care and advance care planning. These themes highlight the multifaceted nature of cancer care in care homes and underscore the importance of addressing residents’ holistic needs in a comprehensive and integrated manner. Improving care home education about cancer and integrating palliative and hospice services within this setting are vital for addressing the diverse needs of residents with cancer.

## 1. Introduction

Cancer exerts a substantial physical, emotional, and financial toll on individuals, families, communities, and healthcare systems worldwide, with many low- and middle-income countries lacking adequate resources to address this burden [1]. The incidence of cancer rises sharply with age due to age-related risk accumulation and the declining efficiency of cellular repair systems [2,3]. Older adults, defined as those aged 65 and older, face a significantly higher risk of developing cancer than younger individuals, with an 11-fold increase in risk [4]. Care homes, nursing homes, and residential facilities that offer personal care and continuous support to older adults requiring assistance with daily living activities are common living arrangements for older individuals with co-morbid conditions, including cancer, dementia, chronic obstructive pulmonary disease, and cardiovascular disease [5,6].

Despite variations in the definition of care homes globally, it is imperative for care home staff to possess the necessary knowledge and competence to manage the care of residents with cancer and collaborate with other healthcare professionals to minimize avoidable transitions to acute care [7,8]. However, caring for care home residents with cancer can be challenging, as care homes often prioritize oncologic management and care to a lesser extent, even though cancer affects up to 26% of their population [9].

Although palliative care and symptom management have gained prominence in non-malignant conditions such as dementia, COPD, frailty, heart failure, and Parkinson’s disease in recent years, cancer may still be overlooked in care homes despite its prevalence in these facilities [10,11,12,13,14]. To enhance the quality of care for individuals living with cancer in care homes, understanding the holistic needs of patients is crucial, as these needs may be intricate and multifaceted. Holistic care entails addressing the whole person, including their social, psychological, emotional, and spiritual needs, in addition to physical health requirements [15].

Therefore, the aim of this review is to synthesize the available evidence on the holistic needs of people living with cancer residing in care home settings. The findings from this review will advance our comprehension of this critical topic and may inform the development of evidence-based interventions and support services tailored to the unique needs of this vulnerable population.

## 2. Materials and Methods

The integrative literature review was conducted using a systematic process following the guidance set out by Whittemare and Knafl [16]. The integrative approach was chosen as the preferred methodology for this review, as it offered a framework for a thorough evaluation, allowing both interventional and non-interventional research to be included, aligning with the review’s goals [16]. Whittemare and Knafl’s [16] five-stage structure was employed, involving problem identification, literature research, data evaluation, data analysis, and presentation. The relevant extracted data were iteratively compared to identify patterns and deviations, enabling a systematic method for data analysis. The four stages of data analysis encompassed data reduction, data display, data comparison, and the formulation and verification of conclusions.

To manage various approaches for data reduction, a classification system was employed for data extraction and coding. For data display, charts were created to facilitate easy comparison of data from primary sources. During the data comparison process, the data displays were repeatedly examined to identify patterns, themes, and connections. More abstraction was used to evaluate patterns, identify commonalities and variances, and subsequently develop conclusions, which were then verified [16]. An audit trail was maintained throughout the process to ensure transparency and traceability. The integrative approach was found to ensure a comprehensive and rigorous analysis, significantly contributing to the fulfilment of the review’s objectives [16].

### 2.1. Search Method

Three databases were selected for the searchers: CINAHLPlus, Medline, and PsychINFO. The selection of databases for this review was based on their comprehensive coverage of medical, psychological, and nursing literature, respectively. These databases collectively provide a well-rounded perspective on the multidimensional aspects of cancer care needs in care homes. Key search terms were developed following an initial review of the literature and were subsequently adapted to align to the primary aim of this integrative review. Key terms were identified in consultation with a subject librarian and wider review team. The search term “holistic needs” was originally searched as a phenomenon of interest; however, this greatly limited the search results. Given the relative paucity of empirical research in the care home setting, the decision was made to search only the population (people living with cancer) and the context (care home settings). The database search involved the following key search terms as MESH terms or free-text terms and were linked together with the AND and OR Boolean operators: (a) cancer* OR oncology OR malignant OR neoplasm* OR tumor OR tumour OR chemotherapy* OR radiotherapy and (b) care home* OR nursing home* OR residential care OR residential home* OR residential care institution* OR long term care facility* OR long-term care setting* OR aged care facility* OR elderly care home* OR palliative care home* OR skilled nursing facility*. To supplement the database searches, reference lists of included papers and grey literature (including the World Health Organization, NHS England, Social Care Online, and Open Grey) were also searched.

### 2.2. Inclusion and Exclusion Criteria

The search strategy was limited to the years 2012–2022 to identify recent research studies. Focusing on a ten-year period, from 2012 to 2022, ensured the inclusion of recent research studies and reflects the evolving landscape of cancer care needs in care homes. This timeframe allowed for the incorporation of up-to-date evidence, considering advancements in medical, psychological, and nursing research over the past decade.

To meet the inclusion criteria for this review, studies had to encompass primary research conducted in care homes with a specific focus on people living with cancer. Additionally, studies that identified cancer as a sub-analysis, even if the main topic was related to other aspects or comorbidities, was also included if it was evident that people living with cancer in care homes were under study. Interventional designs, non-interventional studies, and evidence synthesis were all eligible for inclusion to allow thorough evaluation of the topic. Studies that were not written in English or that did not present empirical data were also excluded.

### 2.3. Data Extraction and Quality Appraisal

Y.C. and G.M. screened the titles and abstracts of all retrieved articles independently and assessed whether they were relevant to the review based on the inclusion and exclusion criteria. A third independent reviewer (S.C.) was appointed to resolve disagreements. Data extraction and a quality appraisal were also conducted by the reviewers (Y.C., G.M., and S.C.). Information on the country, setting, study objective, study design, population, sample size, description of the intervention, outcome measures, results, and relevance were among the pertinent information that was retrieved. Each paper’s methodological rigor was evaluated using the pertinent Joanna Briggs Institute Critical Appraisal Checklist [17]. These structured tools ask a series of questions about the research study’s design and methodology, and the process’s results rank the methodological rigor of studies as weak, moderate, or strong.

### 2.4. Data Analysis

The data analysis was carried out by S.C., Y.T., and G.M., who adopted the narrative synthesis approach [18]. This method involves organizing findings from included studies and describing patterns across them to explore relationships among the data. S.C. utilized a diary to record and reflect on the synthesis process, promoting reflexivity and transparency in the review [18]. The data synthesis occurred in three phases. In the first phase, data extraction involved systematically summarizing each selected study using the JBI Template Source of Evidence Details. This facilitated a preliminary synthesis wherein details of each study were presented in a consistent order, allowing for the identification of potential relationships and differences between them. During the second phase (Phase 2), a thematic analysis of the study findings took place, and finally, in Phase 3, descriptive themes were formulated, revised, and transformed into three overarching themes.

## 3. Results

The search yielded 1099 papers following the removal of duplicates. Following screening of title and abstracts, 30 full-text papers were reviewed. Following full-text review, 20 studies met the inclusion criteria and were included in this review [19,20,21,22,23,24,25,26,27,28,29,30,31,32,33,34,35,36,37,38]. The PRISMA flow chart summarizes the above process (Figure 1).

### 3.1. Characteristics of Included Studies

Twenty primary research studies were included in this literature review, the majority of which were cross-sectional (n = 14), whereas the others were either cohort studies (n = 4) or mixed methods (n = 2). The review included a total sample of 1,384,896 individuals, of which 52% were female, with an average age of approximately 78.5 years. The most common comorbidity among the sample population was neurological or mental health conditions such as dementia, stress, anxiety, or depression (N = 239,228). The studies were conducted in four countries, namely, Norway (n = 9), the USA (n = 9), New Zealand (n = 1), and Canada (n = 1). Three studies were based in long-term care facilities and seventeen studies were conducted in nursing homes.

### 3.2. Methodological Rigor

Of the twenty studies included in the review, fourteen were rated as having strong methodological rigor [20,25,26,27,28,29,30,31,32,33,34,35,36,37] and six as having moderate methodological rigor [19,21,22,23,24,38], as assessed using the Joanna Briggs Institute checklist (2020). Table 1 provides an overview of the characteristics of the included studies.

### 3.3. Synthesis of Evidence

The included studies shed light on the holistic needs of people with cancer residing in care homes. Three main themes emerged from the data: (1) physical needs, which encompassed various aspects of medical care, such as pain management and symptom control; (2) psychological needs, which included social support, emotional well-being, and mental health care; and (3) end-of life care needs, which encompassed a range of issues related to quality of life, such as end-of-life care and advance care planning. These themes highlight the multifaceted nature of cancer care in care homes and underscore the importance of addressing residents’ holistic needs in a comprehensive and integrated manner. These are summarized in Figure 2.

The results and discussion are presented together in this study, employing a narrative synthesis approach [39]. The objective is to offer a cohesive and comprehensive understanding of the research findings while also emphasizing their significance and limitations. Narrative synthesis is a widely recognized methodology for amalgamating evidence from various studies, allowing for the presentation of both quantitative and qualitative findings in a single narrative. Moreover, it facilitates the incorporation of contextual and theoretical perspectives often not captured in traditional reviews. This combined presentation of results and discussion yields a more fluid and integrated portrayal of the findings and their interpretation. To further contextualize and interpret the overall findings, a high-level discussion section is provided at the end of the review.

#### 3.3.1. Theme 1: Physical Needs

The issue of addressing the physical needs of people with cancer in care homes was widely examined in the included studies [25,27,28,30,31,33,34,35,37]. Among these needs, pain management poses significant challenges, especially for those with cancer [27,33]. It has been established that individuals with cancer experience more bodily pain than those without cancer [37]. A study conducted in European long-term care facilities found that nearly half of residents with cancer reported experiencing pain, which supports this finding [9]. Although not statistically significant but relevant in terms of effect size, research by Drageset et al. [25] revealed that people with cancer reported more pain after six years compared to residents without cancer. Furthermore, Blytt et al. [34] and Pimentel et al. [31] reported that care home patients with cancer commonly have untreated pain and that untreated pain can lead to a decline in general health and well-being [22,25,33].

Older adults face various obstacles that may impede their access to pain treatment, including reduced cognition and communication issues [35]. In cases where individuals with severe cognitive impairment cannot effectively communicate their pain, analgesic medication may not be administered [35]. Additionally, other factors contributing to untreated pain in residents with cancer in care homes may include facility-level characteristics and complex care needs of residents, such as the use of feeding tubes or restraints [31]. Proper assessment and treatment of cancer pain are crucial for effective management, and clinicians must have a sufficient understanding of pain to evaluate a patient’s level of pain and prescribe the right drugs according to the World Health Organization’s recommendations [40]. To assess pain, the reviewed studies used the 36-Item Short Form Survey (SF-36) and the Discomfort Behavior Scale (DBS) [25,30,35,37]. The SF-36 questionnaire is a self-administered tool that allows residents to verbally report their pain frequency, which may be less effective for people who are less likely to be prescribed analgesics, including those with cognitive impairment [41]. In contrast, the DBS uses observable indicators to identify pain, which can improve pain management in cognitively impaired residents who do not have verbal abilities [42].

In addition to pain, residents with cancer may have trouble breathing, also known as dyspnea or shortness of breath [19]. However, it has been reported that many people living with cancer in care home settings poorly manage these symptoms [38]. Evidence from Morris and Galicia-Castillo [43] reveals that many patients have dyspnea towards the end of life, and despite national recommendations, healthcare professionals may still be under-recognizing and not managing dyspnea appropriately.

Physical functioning, which includes measures of strength, fitness, and physical activity levels, is closely related to a person’s ability to complete activities of daily living [44]. As people age, their physical functions decline, as highlighted in [34]. However, residents with cancer experience a greater observed decrease in physical functioning compared to residents without cancer [34,37]. This can lead to more complex symptoms and care needs that are difficult to manage and negatively impact the residents’ health-related quality of life (HRQOL) [25]. Physical functioning is mostly subjectively reported using HRQOL questionnaires, and having multiple morbidities and poor health negatively impacts physical functioning [25]. Additionally, undergoing surgery, such as resection of the colon to treat colon cancer, can lead to substantial functional decline [36].

Cancer progression can also result in increased dependence on others for everyday tasks such as personal hygiene and elimination, leading to feelings of powerlessness characterized by a perceived inability to cope with and control one’s own situation [45]. Studies have indicated that staff in care homes often use routines as a means of control and do not take residents’ opinions into account [46].

Overall, this theme on the physical needs of people with cancer in care homes covers pain and its management, breathing difficulties, and reduced physical functioning. Effective management of these key symptoms can address their physical care needs and contribute significantly to their holistic needs and the overall health of the residents.

#### 3.3.2. Theme 2: Psychological Needs

Within the examined studies, a comprehensive theme emerged, highlighting the identification of emotional, social, and spiritual needs. This overarching theme forms the core of a broader exploration into psychological needs.

Considering emotional needs, depression is a widespread phenomenon among people living with cancer residing in care homes. Research has indicated that 55% of cognitively intact nursing home residents with cancer reported experiencing symptoms of depression [24]. In older adults, depression has been linked to various medical disorders and cognitive impairment [47], and in residents with cancer, depression symptoms have been associated with shorter survival time, resulting in a 3.5-month reduction in survival [21].

Altho0ugh depression affects up to 20% of cancer patients and anxiety affects around 10% [48], these care needs are often overlooked in care homes and have only recently received attention [49]. Furthermore, research has revealed that only three out of five caregivers help patients experiencing feelings of worry or melancholy when necessary, indicating that these emotions are the least commonly addressed issues in care homes [38]. Underreporting of anxiety symptoms may be due to general practitioners prescribing medication without formally diagnosing or documenting them in patients’ files [49]. Therefore, it is crucial to increase awareness of anxiety among general practitioners and employees in residential aged care homes and to promote knowledge of available treatment options.

Although depression and anxiety are significant emotional needs, important social needs were also identified. In particular, loneliness is a common concern among people living with cancer who reside in care homes [26]. Studies have indicated that 57% of cognitively intact care home residents with cancer report feeling lonely [23]. Although there is limited research on loneliness among people with cancer, evidence suggests that severe loneliness is generally prevalent among older adults in care homes, ranging from 22% to 42% [50] and from 9% to 81% [51]. Loneliness has been associated with numerous negative health outcomes, such as depression, dementia, cardiovascular disease, malnutrition, low quality of life, and mortality [52]. Despite these risks, loneliness is often not addressed in care homes due to a lack of research and conflicting findings on effective interventions [52]. The COVID-19 pandemic has further highlighted the risks to residents’ social connections and overall health [53].

Residents with cancer also have spiritual needs that require attention. One of the most significant existential issues during the cancer experience is the search for meaning in life and the ways in which meaning is created [54]. An important component of cancer patients’ mental health is their capacity to modify their sense of purpose in life in response to their cancer experience [54]. Many cancer patients report changes in their sense of meaning in life and an increased awareness of life’s limitations, leading them to understand themselves as vulnerable [55]. About 17% of cancer patients report wanting to terminate their lives, not because of pain levels but rather due to melancholy, hopelessness, and meaninglessness [54]. Facilitating meaningful activities is a potential approach to addressing the social and spiritual needs of residents. Engaging in activities that keep residents with cancer engaged during the day can help them find meaning in life [20,24]. For example, in the United Kingdom, the role of well-being and activity coordinators in care homes, regulated by the Health and Social Care Act 2008 (Regulated Activities) Regulations 2014, is crucial in supporting residents in finding meaning in their daily lives. However, this responsibility should not be solely delegated to activity or recreational staff [56,57].

Overall, it is crucial to address the psychological and social needs of cancer patients in care homes, including depression, anxiety, loneliness, the search for meaning, the sense of powerlessness, and communication difficulties. By doing so, we can improve their quality of life and overall well-being. It is essential that healthcare professionals and staff in care homes receive appropriate training to identify and address these needs effectively. Furthermore, more research is needed to understand these issues better and to develop effective interventions.

#### 3.3.3. Theme 3: End-of-Life Needs

Residents with cancer in the end-of-life (EOL) stage require special attention to these care needs. Discussions about death, future care preferences, and hospice services can enable them to have better EOL plans with a greater likelihood of a good death. Although most terminal cancer patients prefer to die at home, they frequently die in hospital or care home settings [38]. This trend has remained unchanged, and there is an increase in cancer deaths in care homes [58]. Timely referral initiated by care home staff members is favored, as uncontrollable pain is often cited as a prevalent cause for admitting patients to hospice [28]. By carefully considering the quality of hospice care options for terminal cancer patients, they can receive high-quality care [38].

In addition, cancer patients in care homes may experience issues with communication and decision-making, as they may face difficulties in conveying their needs and preferences due to cognitive impairment or physical limitations [58]. It is essential that healthcare professionals and staff in care homes work together to support residents in making informed decisions and to involve them in their own care as much as possible [58]. This could involve using communication aids, such as picture boards or symbol charts, or involving family members in discussions about care decisions.

Hospice care has been associated with better management of cancer-related pain. Compared to non-hospice residents, hospice use was related to receiving analgesics, including both scheduled and “as needed” medicine, among residents who self-reported pain [28]. The included studies have found that hospice residents with cancer were more likely to receive opioids for pain relief in the last 2 weeks of life, which is important since opioids are the first-line therapy for moderate to severe cancer pain [30]. Hospice admissions thus appear to be advantageous for long-term care residents [28], as being in hospice may improve the management of pain caused by terminal cancer [30].

In addition, hospice service can help alleviate burdensome transitions at EOL and provide smoother transitions for patients and their families. These transitions, defined as two or more hospital stays or a stay in an intensive care unit within the last 90 days [29], can be stressful and disruptive for patients and their families. Hospice care can help prevent these transitions by providing necessary support and, as a result avoid inconsistency with personal preferences. Many patients at the end of life require emergency department visits for relief of pain or due to breathing difficulties, but with the support of hospice settings these visits are often unnecessary [32]. Furthermore, hospice staff have specialized training and expertise in pain and symptom management, making it easier to manage pain–even in new residents they have never seen before—compared to care home staff [28].

Although this review suggests better outcomes for care home residents who move to hospice services, there are several limitations to this approach. As a person’s home, care homes are often the place where residents feel most comfortable and supported. The staff in care homes develop relationships with the residents over time and are more familiar with their needs and preferences [59]. Thus, allowing care home residents to remain in familiar surroundings can provide a sense of continuity, familiarity, and comfort in their final days. Disruptive moves to hospices may cause significant stress and anxiety for residents and their families. Hospices, despite being staffed with experts in end-of-life care, may not be able to provide the same level of individualized care as care homes due to the lack of knowledge about residents’ medical histories and preferences [60]. This lack of familiarity may lead to a less personalized care experience, which can be distressing for both the residents and their loved ones. Furthermore, good deaths can and do occur in care homes [61]. By offering effective end-of-life care services in care homes, residents can receive care tailored to their individual needs and preferences without the disruption of moving to a hospice care facility. Additionally, hospice staff can collaborate with care home staff to provide the necessary support and training to ensure high-quality end-of-life care for residents.

In summary, facilitating care home residents living with cancer to remain in familiar surroundings can provide a sense of continuity, familiarity, and comfort in their final days. Disruptive moves to hospice care facilities may cause significant stress and anxiety, and good deaths can and do occur in care homes with appropriate palliative care services.

## 4. Discussion

Although this review has identified some areas of care needs for people with cancer living in care homes, other areas like cancer diagnosis and spirituality appear to be underreported. Cancer diagnosis can cause significant spiritual distress, and spiritual needs should be recognized, understood, and considered when caring for patients [62]. Furthermore, spirituality is a complex and multifaceted concept, particularly concerning differentiating between religion and spirituality in non-religious individuals [63]. Effective tools and procedures for assessing spiritual needs and providing appropriate interventions at an early stage should be developed [64]. Research has shown that there is a positive association between spiritual needs, religious coping, and quality of life (QoL) domains in cancer patients, indicating a need for greater attention to this area of care [65]. These represent key areas or priorities for future research.

Residents with cancer living in care homes are more likely to live with other diseases such as dementia, stroke, cardiovascular disease, or diabetes. This population presents with high prevalence of multi-morbidity, frailty, and geriatric syndromes, which require complex management [66]. Cancer management and care are further complicated by medical procedures, treatment-related toxicity, supportive care needs, polypharmacy, and depression, which can also lead to functional and cognitive decline [67]. Unsurprisingly, this review has shown that residents living with cancer in care home settings are more likely to experience lower health-related quality of life compared to those with no history of cancer due to comorbidity burden, physical and mental symptoms, and treatment-related issues. It also seems probable that the issue is exacerbated by the lack of educational or training initiatives available to aid personnel in enhancing the quality of care provided to residents with cancer [68].

Contextually, the SARS-CoV-2 pandemic continues to have a significant impact on long-term care facilities, where rates of infection among staff and residents are among the highest and account for a considerable proportion of COVID-19 deaths [69,70]. Workforce issues, such as staff shortages, high turnover, and inadequate pay and training for direct care workers, have also been highlighted [59]. Addressing these problems requires bridging the gap between nurse staffing and residents’ needs in long-term care facilities, as research has shown that the quality of care is correlated with nurse staffing levels [71].

Effective cancer care for individuals in care homes requires a collaborative approach involving the person with cancer, their family, and a care team consisting of multiple healthcare professionals from different disciplines. The Enhanced Health in Care Homes Framework [72] recommends a multi-disciplinary team (MDT) approach to provide coordinated and comprehensive care for residents with cancer. Nurses play a key role in promoting partnership between care providers, which can have positive outcomes for patients [73]. Advance care planning (ACP) is also a critical aspect of multidisciplinary practices, particularly as care homes are becoming increasingly more common as the last residence for older adults. ACP is a process that allows individuals to make anticipatory decisions about their care and should be facilitated in care homes for all people living with cancer. The qualifications and capabilities of staff significantly impact the quality of care in care home settings, particularly in undertaking ACP, administering medication, or delivering palliative and end-of-life care. Insufficient training in medication administration among care home staff [38] significantly undermines the quality of care, especially in facilities with limited staffing, consequently impacting the care of residents in care settings [74]. However, a substantial body of evidence suggests that education and training can equip staff to improve resident outcomes. For instance, targeted training initiatives have shown success in fostering the adoption of advance care planning practices [75] and enhancing support provided to family members in care home settings [76]. These interventions not only address immediate challenges but also contribute to an overall improved standard of care within care home settings. 

Although the review addresses the holistic needs of residents with cancer in care homes, it is important to acknowledge the exclusion of certain essential aspects in the existing literature. Notably absent are discussions on practical concerns such as financial challenges, legal issues, and housekeeping matters, which can significantly impact the well-being of residents. Additionally, knowledge-related concerns pertaining to disease awareness, treatment procedures, and available services are underrepresented in the current body of literature. These omissions underscore the importance of considering a broader spectrum of factors that contribute to the comprehensive care of individuals with cancer in care home settings. Future research should strive to encompass these often-overlooked dimensions to provide a more nuanced understanding of the multifaceted needs of this population.

Overall, this review offers an in-depth discussion of the holistic needs of people with cancer living in care homes. These individuals often have complex medical histories, and their care needs may not always be fully acknowledged, particularly in terms of psychological, physical, and psychosocial needs.

### 4.1. Implications for Practice

The literature suggests that not all residents with cancer automatically require end-of-life care, and some may need support to adapt to living with cancer for an extended period. Some authors recommend upskilling care home nurses to deliver a higher or more specialized level of cancer care to residents. Incorporating palliative and hospice care services in care home settings can be a beneficial approach to addressing the holistic needs of residents with cancer. Hospice care teams, comprising medical professionals, nurses, social workers, chaplains, and volunteers, collaborate to provide comprehensive care and support, encompassing the physical, emotional, and spiritual needs of residents.

This review has demonstrated that hospice care can significantly alleviate symptoms commonly associated with cancer, such as pain and breathlessness. Furthermore, trained professionals offer emotional and spiritual support to residents and their families during the end-of-life phase. Additionally, hospice care aids patients and their families in making decisions about end-of-life care and provides support during the dying process. Given their specialized knowledge of their residents, care home nurses, when equipped with the necessary tools and resources, can play an integral role in delivering high-quality care to residents with cancer, empowering them to meet their unique needs effectively.

### 4.2. Strengths and Limitations

This integrative review draws upon diverse study designs, facilitating a comprehensive synthesis of data from multiple sources. The review was conducted following a robust framework set out by experts in the field of integrative reviews, ensuring rigorous inclusion and exclusion criteria. The methodology employed for this integrative review enabled a thorough analysis of the available literature, with a specific emphasis on identifying overarching themes and patterns. However, it is important to acknowledge certain limitations in this review. One potential limitation is that all included studies were published in English, meaning that key literature published in other languages has not been included. A further limitation is that some of the studies included in this review were conducted by the same team of researchers on the same sample population, which may restrict the diversity of the population studied, possibly affecting the generalizability of the findings. To mitigate potential biases and enhance the review’s reliability, several strategies were implemented. For instance, two researchers independently conducted screening, data extraction, and data interpretation, with any discrepancies resolved through triangulation with a third researcher. Additionally, the overall team provided feedback, contributing to the review’s robustness.

Despite these limitations, this integrative review followed a rigorous methodology for data extraction and synthesis, striving to provide valuable insights into the topic under investigation.

## 5. Conclusions

This integrative review has illuminated the holistic needs of people living with cancer residing in care home settings, emphasizing the significance of comprehensive care to address their physical, psychological, and end-of-life requirements. The review’s methodology, following a robust framework, allowed for the synthesis of data from diverse sources, leading to a comprehensive analysis of the available literature. Moving forward, it is crucial for care home staff, particularly nurses, to upskill in providing specialized cancer care, and the incorporation of palliative and hospice care services in care home settings can be instrumental in meeting the diverse needs of residents with cancer. By empowering care home nurses with adequate tools and resources, they can play a pivotal role in delivering high-quality, patient-centered care to this vulnerable population.

## Figures and Tables

**Figure 1 healthcare-11-03166-f001:**
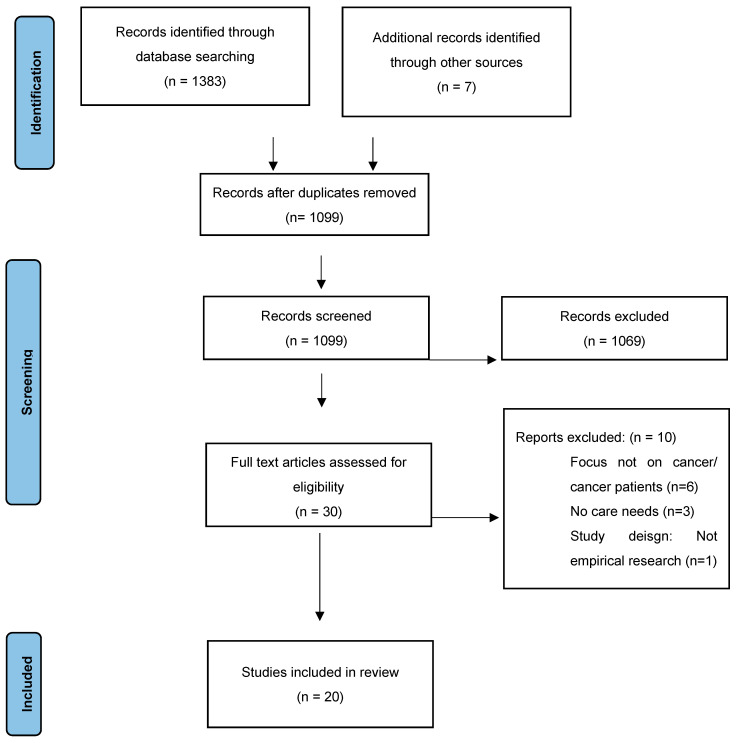
PRISMA flow diagram. PRISMA, Preferred Reporting Items for Systematic Reviews and Meta-Analyses.

**Figure 2 healthcare-11-03166-f002:**
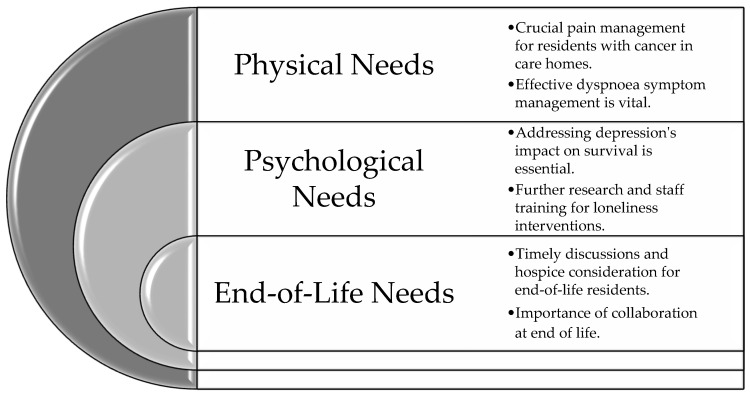
Summary of findings.

**Table 1 healthcare-11-03166-t001:** Characteristics of studies included in this review.

Authors, Country	Aim of Research	Research Design	Outcome of Critical Appraisal (JBI, 2020)	Sample and Setting	Main Findings
Bainbridge et al., 2015 Canada [32]	To examine the contribution of covariates to having an emergency department (ED) visit in the last 6 months of life or to dying in hospital.	Cohort study	Strong	Long-term care facility residents (n = 1196) Long-term care facility	A total of 61% visited the emergency department in the last 6 months of life (average of 2.3 visits per person) and 20% died in the hospital. Cancer type, income, gender, time in long-term care, and rural location did not predict acute care outcomes. However, comorbidities, younger age, and region of residence were significant predictors of emergency department visits and/or hospital deaths.
Blytt et al., 2018 Norway [35]	To identify the prevalence of cancer and differences regarding neuropsychiatric symptoms (NPSs) and medication among nursing home (NH) patients with and without dementia and cancer.	Cross-sectional study	Strong	Nursing home residents (n = 1825) with and without dementia with cancer from 64 nursing homes in Norway. Nursing homes	Patients with comorbid dementia and cancer received significantly more analgesics than patients without cancer but with dementia. Patients with comorbid dementia and cancer also had significantly more NPS, including sleep disturbances and agitation, compared to patients without dementia but with cancer.
Boyd et al., 2019 New Zealand [19]	To describe the end-of-life experience of those living in LTC facilities in New Zealand.	Cross-sectional study	Moderate	Residents in 61 long-term care facilities (n = 286) who had died from cancer (17%), dementia (49%), both cancer and dementia (4%), or another chronic illness (30%). Long-term care facility	Palliative care principles should be integrated with geriatric care to provide high-quality end-of-life care in LTC facilities. It is essential that those working in LTC facilities recognize palliative care philosophy, and specialist palliative care providers work collaboratively with gerontologists to ensure high-quality end-of-life care for people with complex geriatric syndromes.
Drageset et al., 2020 Norway [26]	To examine loneliness among nursing home residents over 6 years and whether sociodemographic factors, sense of coherence, social support, or depression symptoms might influence loneliness.	Cross-sectional study	Strong	Nursing home residents (n = 122) living with cancer aged 65 years or older. Nursing homes	This study has three important findings. First, loneliness did not change over time during the 6 years of follow-up. Second, symptoms of depression and the sense of coherence seem to be important components of loneliness. Third, social support dimensions and having a diagnosis of cancer were not associated with loneliness.
Drageset et al., 2012 Norway [37]	To study the sociodemographic characteristics and HRQOL among NH residents with and without a cancer diagnosis, adjusting for comorbidity.	Cross-sectional study	Strong	Nursing home residents (n = 60) living with cancer aged 65 years or older. Nursing homes	NH residents with cancer reported worse HRQOL compared to those without cancer, including more pain and worse general health. However, after adjusting for comorbidities, the difference in general health was no longer statistically significant. Cognitively intact residents with cancer reported less role limitation related to emotional problems.
Drageset et al., 2013 Norway [22]	To investigate loneliness and mortality among cognitively intact NH residents with cancer vs. those without cancer.	Cross-sectional study	Moderate	Nursing home residents (n = 227) with cognition intact, with cancer (n = 60), and without cancer (n = 167) from 30 nursing homes. Nursing homes	Age, education, comorbidity, and emotional loneliness (attachment) were associated with mortality in nursing home residents, but survival did not differ significantly between residents with and without cancer.
Drageset et al., 2013 Norway [21]	To investigate whether anxiety and/or depression: 1. Are associated with survival and 2. Have different effects on survival; for residents with and without cancer.	Cross-sectional study	Moderate	Nursing home residents (n = 227) with cognition intact, with cancer (n = 60), and without cance4r (n = 167) from 30 nursing homes. Nursing homes	Depression and comorbidity predicted 5-year mortality for NH residents. Cancer diagnosis did not have a significant impact on survival time, but anxiety symptoms predicted shorter survival for residents with cancer. Caregivers should closely observe residents with cancer for anxiety symptoms and depression and comorbidity should be monitored regardless of cancer diagnosis.
Drageset et al., 2015 Norway [23]	To investigate loneliness and social support among cognitively intact nursing home residents with cancer.	Mixed-methods study	Strong	Nursing home residents (n = 60) living with cancer aged 65 years or older. Nursing homes	Loneliness is associated with reassurance of worth. It is experienced as inner pain, loss, and feeling small. To alleviate loneliness, one should be engaged in activities, be in contact with others, and occupy oneself.
Drageset et al., 2015 Norway [20]	To investigate the holistic needs of cancer patients living in care homes without a cognitive impairment during an acute hospital admission.	Cross-sectional study	Strong	Nursing home residents (n = 60) living with cancer aged 65 years or older. Nursing homes	Residents with cancer had more admissions (25/60) than those without (53/167). Social integration was correlated with admission (*p* = 0.04) regardless of cancer diagnosis.
Drageset et al., 2016 Norway [24]	To investigate the sense of coherence (SOC) and depression among cognitively intact NH residents with cancer and their experience with depression and coping.	Mixed-methods study	Moderate	Nursing home residents (n = 60) living with cancer aged 65 years or older. Nursing homes	This study found that the General Depression Scale (GDS) was significantly correlated with SOC, with a stronger SOC associated with less symptoms of depression. The experience of sadness was identified as a dominant theme in coping with symptoms of depression. Over half of nursing home residents reported symptoms of depression, highlighting the need to pay attention to their experience of depression and SOC to improve their situation.
Drageset et al., 2017 Norway [25]	The study aimed to examine HRQOL over time during a 6-year period among residents of NHs who are not cognitively impaired and to examine whether sense of coherence and a diagnosis of cancer influence HRQOL.	Cross-sectional study	Strong	Nursing home residents (n = 227) with cognition intact. Nursing homes	During a 6-year follow-up, physical functioning and role limitation-physical sub-scores declined, and having cancer at baseline was negatively associated with general health. Sense of coherence at baseline was positively correlated with all sub-scores of a 36-Item Short Form Survey (SF-36). Sense of coherence was found to be an important component of HRQOL, and having cancer was linked to a decline in general health.
Dubé et al., 2018 USA [27]	To evaluate whether the documentation and management of pain varies by level of cognitive impairment among nursing home residents with cancer.	Cross-sectional study	Strong	Nursing home residents (n = 367, 462) with cancer. Nursing homes	For those with staff-assessed pain, pain prevalence was 55.5% with no/mild cognitive impairment and 50.5% in those severely impaired. Pain was common in those able to self-report. Greater cognitive impairment was associated with reduced prevalence of any pain. Pharmacologic pain management was less prevalent in those with severe cognitive impairment.
Finlayson et al., 2012 USA [36]	To compare outcomes of nursing home residents undergoing colectomy with benchmark mortality and functional decline in the general nursing home population.	Cohort study	Strong	Nursing home residents (n = 6822) who underwent surgery for colon cancer who were 65 years or older. Nursing homes	Nursing home residents who underwent colectomy experienced a 3.9-point worsening in Minimum Data Set (MDS)-ADL score on average after 1 year, with mortality and sustained functional decline rates of 53% and 24%, respectively. Older age, readmission after surgical hospitalization, and surgical complications were associated with functional decline at 1 year. The study suggests that initiatives aimed at improving surgical outcomes are needed in this vulnerable population.
Hunnicutt et al., 2017 USA [28]	To estimate the extent to which receipt of hospice in nursing homes (NHs) increases the receipt of pain management for residents with cancer at the end of life.	Cohort study	Strong	Medicare beneficiaries with cancer (n = 78,160) who were nursing home residents in the last 90 days of life. Nursing homes	In hospice residents, pain prevalence was higher compared to non-hospice residents, but untreated pain was uncommon. Hospice use was associated with receiving scheduled analgesics and medication “as needed”, and this association was similar in residents with staff-assessed pain. Therefore, hospice is linked with increased pain management in those with documented pain.
Lage et al., 2020 USA [29]	To examine factors associated with potentially burdensome end-of-life (EOL) transitions between care settings among older adults with advanced cancer in nursing homes (NHs).	Cohort study	Strong	Deceased older nursing home residents with cancer (n = 34,670). Nursing Hhomes	A study of 34,670 subjects showed that 53.8% had moderate to severe cognitive impairment and full dependence on activities of daily living (ADLs). A total of 56.3% of patients used hospice in the 90 days before death, while 36% had potentially burdensome end-of-life transitions, which was higher in patients who did not receive hospice care. Factors associated with higher risk of burdensome EOL transitions were full dependence on ADLs, congestive heart failure, and chronic obstructive pulmonary disease. Patients with do-not-resuscitate directives and impaired cognition had lower odds of burdensome EOL transitions.
Mack et al., 2018 USA [33]	To evaluate racial disparities in pain management among residents with cancer in nursing homes at time of admission	Cross-sectional study	Strong	Newly admitted nursing home residents with cancer (n = 342,920) Nursing homes	Among nursing home residents with cancer, 60% reported pain, with non-Hispanic Blacks less likely to have both self-reported pain and staff-reported pain documentation compared with Non-Hispanic Whites. Although most residents received some pharmacologic pain management, Blacks were less likely to receive any compared with Whites, consistent with differences in receipt of non-pharmacologic treatments.
Monroe et al., 2012 USA [35]	To assess pain in nursing home residents with mild to very severe Alzheimer’s disease (AD) who died from cancer.	Cross-sectional study	Strong	Nursing home residents (n = 48) with mild to severe dementia Nursing homes	The severity of Alzheimer’s disease was negatively related to pain behaviours, with a significant difference between moderate and very severe Alzheimer’s. There was no significant correlation between opioid analgesics and pain behavior, but individuals with severe Alzheimer’s disease received fewer opioids.
Monroe et al., 2013 USA [30]	The aim of the current pilot study was to examine the association between hospice enrolment, dementia severity, and pain among nursing home residents who died from advanced cancer.	Cross-sectional study	Strong	Nursing home residents (n = 55) with dementia who died from cancer, 54.5% female. Mean age 86.4 ± 7.8. Nursing homes	About 45% of nursing home residents were in hospice at the end of life, and they were more likely to receive an opioid but less likely to have severe cognitive impairment. Hospice enrolment was linked to an increased chance of receiving an opioid, but lower cognitive functioning decreased the likelihood of receiving an opioid. However, it is noteworthy that 40% of nursing home residents with dementia who died from cancer did not receive any opioid during this time.
Parast et al., 2021 USA [38]	To compare the quality of hospice care provided in various venues while looking at the experiences of decedents with a primary cancer diagnosis and their family caregivers.	Cross-sectional study	Moderate	Caregiver respondents (n = 217,596) whose family member had a primary cancer diagnosis and died in 2017–2018 while receiving hospice care from 2890 hospices. Long-term care facilities	The study found that quality measures for care in nursing homes and assisted living facilities varied significantly, with scores ranging from 74.9 for receiving hospice care training to 89.5 for treating family members with respect. Caregivers of deceased patients consistently reported lower quality of care, with scores varying significantly across settings. The overall score for obtaining treatment for symptoms was 75.1, with scores ranging from 60.6 to 84.5 for individual items within the measure.
Pimentel et al., 2015 USA [31]	To assess improvements in pain management of NH residents with cancer since the implementation of pain management quality indicators.	Cross-Sectional Study	Strong	Nursing home residents (n = 8094) with cancer. Nursing Homes	The study found that over 65% of nursing home residents with cancer had some level of pain, with 28.3% experiencing daily pain and 37.3% experiencing pain less than daily. Of those with pain, 13.5% had severe pain and 61.3% had moderate pain. Women, residents admitted from acute care or who were bedfast, and those with compromised activities of daily living, depressed mood, indwelling catheter, or terminal prognosis were more likely to have pain. However, over 17% of residents with daily pain received no analgesics, including 11.7% with daily severe pain and 16.9% with daily moderate pain. The treatment was negatively associated with age > 85 years, cognitive impairment, presence of a feeding tube, and restraints.

## Data Availability

The datasets generated and/or analyzed during the current study are available from the corresponding author upon reasonable request.

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
