# Peer review of "Exploring the Holistic Needs of People Living with Cancer in Care Homes: An Integrative Review"

_healthcare, 2023, doi:10.3390/healthcare11243166_

Round 1

Reviewer 1 Report

Comments and Suggestions for Authors

The authors aim to review and summarize key studies between 2012-2022 exploring the holistic needs of people living with cancer in care homes. The authors have also added commentary on the summary of evidence thus far, clinical implications and strengths/limitations of the current work, adding further value to the manuscript.

Overall, this is a clearly written, comprehensive manuscript, significant to pertinent clinical, research, policy driving readership, and some sections of the general audience.

The reviewer recommends minor edits and feedback prior to acceptance for publication -

1.     It would be helpful to summarize the 3 described themes (section 3.3. Synthesis of evidence) in a detailed figure. This would be helpful to the manuscript.

2.     Section 3.3.2. Theme 2: This section has subsets that define emotional, social and spiritual needs. Suggest rearranging and describing this section within the context of these parameters.

3.       Section 4. Discussion - would recommend adding a summary of some other factors that have not been covered in the studies included for this manuscript: practical concerns (finances, legal issues, housekeeping, etc.), knowledge concerns (disease, treatment/procedures, service, etc.).

Comments on the Quality of English Language

 Overall, the quality of English is good.

Please recheck text for typos, e.g., section 2.1. Search Method – line 95, 96, 98 should be ‘neoplasia’, ‘chemotherapy’, ‘facility’.

Author Response

The reviewer recommends minor edits and feedback prior to acceptance for publication -

  1. It would be helpful to summarize the 3 described themes (section 3.3. Synthesis of evidence) in a detailed figure. This would be helpful to the manuscript.

Thank you.  We have now included figure two within section 3.3.

  1. Section 3.3.2. Theme 2: This section has subsets that define emotional, social and spiritual needs. Suggest rearranging and describing this section within the context of these parameters.

 Thank you for this feedback.  We have highlighted changes to 3.3.2 as recommended and feel this theme is more clearly represented.

  1.      Section 4. Discussion - would recommend adding a summary of some other factors that have not been covered in the studies included for this manuscript: practical concerns (finances, legal issues, housekeeping, etc.), knowledge concerns (disease, treatment/procedures, service, etc.).

Thank you for this helpful feedback.  We have included a paragraph at the conclusion of the discussion that strengthens our work.

Comments on the Quality of English Language 

 Overall, the quality of English is good.

Please recheck text for typos, e.g., section 2.1. Search Method – line 95, 96, 98 should be ‘neoplasia’, ‘chemotherapy’, ‘facility’.

Thank you.  We have used the asterisk symbol for truncation.

Reviewer 2 Report

Comments and Suggestions for Authors

I read the review with great interest. The review generally presents the needs of people in care homes.

I would suggest that the following minor recommendations be considered:

-          In the abstract the findings of review are lacking

-          The aim of the review should be highlighted at the end of the introduction

-          Line 387 – "A lack of staff…" – this is the main problem in many countries and could be discussed in more detail

Author Response

I would suggest that the following minor recommendations be considered:

-          In the abstract the findings of review are lacking

Thank you.  We have rewritten the end of the abstract to summarise the themes and provide more compelling conclusion.

-          The aim of the review should be highlighted at the end of the introduction

Thank you.  We have explicitly added the aim of the review to the beginning of our final paragraph in section 1.

-          Line 387 – "A lack of staff…" – this is the main problem in many countries and could be discussed in more detail

Thank you.  We have extended this section as a result of this helpful feedback.

Reviewer 3 Report

Comments and Suggestions for Authors

The manuscript contains significant information to justify publication.

The title is consistent with the other items of the manuscript (objective, results, and conclusion) and provides key information about the main objective the review addresses.

The Summary presents sufficient information to understand the procedures performed and their outcome.

The Introduction is presented clearly and in a logical sequence. Explains the concepts used and the justification for the study,

The Method is consistent with the title and objective and provides a complete description of the methodological procedures. The authors describe the search strategies used and how the studies found in the search were analyzed to include articles that answered the research question.

In line 111, the authors present the exclusion criteria, but it is not appropriate for the exclusion criteria to be the opposite of the inclusion criteria. For example: inclusion criteria: research conducted in care homes with a specific focus on people living with cancer and exclusion criteria: studies that were not conducted in care homes or did not focus on people living with cancer. One is the exact opposite of the other. I suggest removing these specific exclusion criteria.

In the diagram presented, I suggest that authors present the number of articles obtained from searches in each of the databases and not just the total.

The Results were perfectly presented.

In the Discussion, the authors emphasize important aspects observed in the studies included and discuss agreements and divergences with other published research. They present the limitations and the contributions of the study.

The Conclusions respond to the proposed objective.

Author Response

Reviewer 3:

In line 111, the authors present the exclusion criteria, but it is not appropriate for the exclusion criteria to be the opposite of the inclusion criteria. For example: inclusion criteria: research conducted in care homes with a specific focus on people living with cancer and exclusion criteria: studies that were not conducted in care homes or did not focus on people living with cancer. One is the exact opposite of the other. I suggest removing these specific exclusion criteria.

Thank you.  We have deleted the exclusion criteria and this reads better.

In the diagram presented, I suggest that authors present the number of articles obtained from searches in each of the databases and not just the total.

Thank you for this feedback.  We have not included this information in the diagram as we do not want to confuse the reader (there are a high number of duplicates) and many of the identified texts were found across multiple databases.

Reviewer 4 Report

Comments and Suggestions for Authors

An integrative review was conducted to explore the holistic needs of people living with cancer in care homes. The primary objective of this integrative literature review was to consolidate the available evidence concerning the comprehensive needs of people living with cancer in care home settings, providing valuable insights into addressing their diverse needs.

The objective was met and a detailed discussion was written. The limitations and implications for practice were discussed.

- Is there any specific reason for choosing these 3 databases "CINAHLPlus, Medline and Psych-INFO"?

- Was there publication bias in your review, since you excluded studies published in other languages? 

Author Response

- Is there any specific reason for choosing these 3 databases "CINAHLPlus, Medline and Psych-INFO"?

Thank you.  We have included rationale to this in the main body.

- Was there publication bias in your review, since you excluded studies published in other languages? 

Thank you.  We have explicitly added this to our 4.2 strengths and limitations section.